# Associations between Coping Profile and Work Performance in a Cohort of Japanese Employees

**DOI:** 10.3390/ijerph19084806

**Published:** 2022-04-15

**Authors:** Yuichiro Otsuka, Osamu Itani, Yuuki Matsumoto, Yoshitaka Kaneita

**Affiliations:** Division of Public Health, Department of Social Medicine, Nihon University School of Medicine, 30-1 Oyaguchi-kamimachi, Itabasi-ku, Tokyo 173-8610, Japan; itani.osamu@nihon-u.ac.jp (O.I.); matsumoto.yuuki78@nihon-u.ac.jp (Y.M.); kaneita.yoshitaka@nihon-u.ac.jp (Y.K.)

**Keywords:** coping profiles, cohort study, Japan, work stress, productivity

## Abstract

This study aimed to investigate the effects of coping profiles on work performance. Data were collected during a 2-year prospective cohort study of 1359 employees in Japan. Participants completed a self-administered questionnaire in 2018 (T1; baseline) and again in 2020 (T2; followup; followup rate: 69.8%) to enable the assessment of work performance, perceived stress, and stress coping profiles at T1 and T2. Multivariate logistic regression models and causal mediation analysis were performed to identify the effects of coping profiles on work performance. Covariates included age, sex, company, job type, employment status, working hours, holidays, and lifestyle behaviors (e.g., smoking, sleep duration). A dysfunctional coping profile (β = −1.17 [95% CI, −2.28 to −0.06], *p* = 0.039) was negatively associated with work performance. Coping profiles of planning (β = 0.86 [95% CI, 0.07–1.66]) and self-blame (β = −1.33 [95% CI, −1.96 to −0.70], *p* < 0.001) were significantly associated with work performance. Dysfunctional coping, specifically, self-blame (β = −1.22 [95% CI, −1.83 to −0.61]), mediated the association between stress and work performance. Thus, some coping profiles may lead to an increase or decrease in work performance. The possible impact of coping strategies on workers’ productivity requires further exploration. Furthermore, information on effective coping profiles should be incorporated into occupational health examinations.

## 1. Introduction

In recent decades, managers and organizations have both been concerned about the increase in employee health costs [1]. Employee health costs include the direct cost of health plans, as well as costs associated with employee absenteeism and reduced productivity [2]. Impaired work productivity includes lost work time owing to the habitual absence of employees from work (absenteeism) and their reduced work performance while working (presenteeism) [3]. In fact, previous studies have reported that presenteeism is associated with many diseases, such as allergies [3], arthritis [3], depression [4], back pain [4], and hypertension [5]. The key determinants of presenteeism were reported to be stress in the workplace [6], health conditions [7], work–life imbalance [8], and individual factors [9] such as personality and family problems [10]. Additionally, presenteeism often leads to future absenteeism [11]. Thus, presenteeism, in comparison to absenteeism, may be a more important predictor to consider when examining workers’ health. 

Stress in the workplace is an inevitable part of work because of components of the work environment, such as high job demand, poor work management, and negative social climate [12]. A cross-sectional study of 1440 German employees suggested that interventions aimed at reducing job stress have a strong potential for improving presenteeism [13]. Occupational stress is not only related to work performance but also influences physical and psychological health [14]. According to the Transactional Stress Model [15], coping strategies are vital elements in the stress process because they could help mitigate the impact of stressors on health. Coping strategies are categorized by how individuals react to or handle stress. Effective coping strategies lead to faster resolution of privations and help maintain individual health during stress, resulting in a greater sense of safety and security [16]. Ineffective coping strategies, on the other hand, lead to poor physical and mental health, prevent people from avoiding difficult situations, and prolong negative feelings [16]. For example, a cross-sectional study of 1277 U.S. employees showed that adaptive coping strategies may be more effective than maladaptive coping strategies in terms of perceived stress management. A research framework to improve workplace health suggests implementing interventions to help reduce work stress and adopt effective coping strategies [17].

When encountering stressful situations, individuals employ a combination of varied coping strategies [18]. The coping strategies can be categorized into three to four types. COPE and its shortened version, Brief-COPE, two of the most commonly used coping scales, have suggested some common patterns of strategies called coping profiles [19,20]. For instance, some individuals tend to use a combination of problem-focused and emotion-focused coping strategies, whereas some individuals tend to employ primarily dysfunctional coping strategies, and others use very few strategies to deal with life’s stressors [21,22,23].

A few previous studies have shown that specific coping profiles of employees are associated with work performance [24,25,26,27]. For example, a cross-sectional study of 161 U.S. workers showed that a problem-focused coping profile had a positive direct effect on work performance [24]. In addition, coping profiles were reported as mediators between stress and work performance. For example, a short cohort study of 68 Israeli soldiers revealed an emotion-focused coping profile at baseline that predicted low performance at follow-up [25]. A cross-sectional study in Taiwan reported that nurses with emotion-focused and dysfunctional coping profiles had lower levels of job satisfaction [27]. Similarly, a cross-sectional study of 554 Israeli workers demonstrated that coping profiles, such as withdrawal, were negatively associated with performance [26]. However, these studies had some limitations, such as small sample size, a lack of valid coping scales, and limited longitudinal investigations. Further, although previous studies have focused on the relationship between stress coping and stress, and the relationship between stress coping and work productivity, few have investigated the interactions among these three factors.

To address these gaps, we conducted a longitudinal study focusing on general Japanese employees to survey how coping profiles would affect worker performance (i.e., presenteeism). Most studies have shown that distress directly influences work performance [13,14]. However, little is known about whether coping profiles affect performance independently of stress or whether they mediate the relationship between stress and performance. Therefore, we formulated the following two hypotheses: First, a problem-focused coping profile would positively affect work performance, whereas emotion-focused and dysfunctional coping profiles would negatively affect work performance (Figure 1, H1). Second, coping profiles may mediate the relationship between perceived stress and work performance. Thus, we examined the potential mediating role of coping profiles in the prospective relationship between perceived stress at baseline and work performance at follow-up. We also aimed to determine the proportion explained by the mediating effect of each coping profile between them (Figure 1, H2).

## 2. Materials and Methods

### 2.1. Design and Participants

A 2-year prospective cohort study was conducted between May and August 2018 and September and November 2020 with workers employed at six companies in Japan. The companies comprised those from the fields of production, information technology, medicine, and precision equipment. Eligible participants completed a self-administered questionnaire survey. The recommendations of the Strengthening the Reporting of Observational Studies in Epidemiology (STROBE) statement for observational studies were followed [28]. All workers provided written informed consent to participate in the study. This study was conducted according to the guidelines of the Declaration of Helsinki, and it was approved by the ethics committee of the Nihon University School of Medicine (No. 29-12-0).

### 2.2. Measures

Employees completed a self-administered questionnaire that sought details regarding age, sex, job categories (manager and others), number of holidays per month, working time per week, types of coping profiles, lifestyle habits, and perceived stress at baseline (T1). Presenteeism (work performance) was assessed at baseline (T1) and at followup (T2).

#### 2.2.1. Presenteeism (Work Performance)

The Japanese version of the World Health Organization Health and Work Performance Questionnaire (WHO-HPQ) short form [29] was used to assess work performance at baseline (T1) and at followup (T2). The WHO-HPQ includes two aspects: absolute presenteeism and relative presenteeism. Absolute presenteeism indicates actual performance, whereas relative presenteeism is the ratio of actual performance compared to that of other workers in the same job at the same office [30]. In this study, we used absolute presenteeism as a measure of work performance. Absolute presenteeism involves a single-item questionnaire, assessing presenteeism over the past four weeks through the following question: “On a scale from 0 to 10, where 0 is the worst job performance anyone could have at your job and 10 is the performance of a top worker, how would you rate your overall job performance on the days you worked during the past four weeks?” The final score was obtained by multiplying the respondent’s answer by 10 (a range of 0–100). Lower scores indicate a low level of work performance. The test–retest correlation of the WHO-HPQ in this study was 0.31. 

#### 2.2.2. Coping Profile: The Brief-COPE

The Brief Coping Orientation to Problems Experienced (Brief-COPE) was used to evaluate coping profiles. This scale consists of 28 items that measure 14 different types of coping profile, including active coping, planning, positive reframing, acceptance, humor, religion, emotional support, instrumental support, self-distraction, denial, venting, substance use, behavioral disengagement, and self-blame [20]. Each item was rated on a 4-point Likert scale, and higher scores indicate more frequent use of a coping category. Copper et al. described the 14 subscales of the Brief-COPE as reflecting predominantly problem-focused (active coping, instrumental support, planning), emotion-focused (acceptance, emotional support, humor, positive reframing, and religion), and dysfunctional coping profiles (behavioral disengagement, denial, self-distraction, self-blame, substance use, and venting) [31,32]. Higher scores indicate more frequent use of a coping category. To aid in the interpretation of regression results, scores for each coping subscale were expressed as z-scores, meaning these three coping subscales converted a mean of 0 and a standard deviation of 1. Cronbach’s α for the total Brief-COPE in the present study was 0.81. Cronbach’s alpha values for the problem-focused, emotion-focused, and dysfunctional coping subscales were 0.71, 0.67, and 0.64, respectively.

#### 2.2.3. Perception of Stress

The 10-item Perceived Stress Scale (PSS-10) was used to measure the perception of stress [33]. The PSS-10 assesses the degree to which life events in the previous 30 days are perceived as stressful [33]. The 10 items ask about feelings and thoughts that evaluate the degree to which respondents perceive their current life situation as unpredictable, uncontrollable, and stressful. Each item is rated on a 5-point Likert scale. Total scores range from 0 to 40, and the higher the score, the higher the perceived stress levels [33]. The Japanese version of the PSS-10 has been validated [34]. Cronbach’s α for the PSS-10 in the present study was 0.84. 

### 2.3. Covariates

Covariates were selected based on logical, theoretical, and/or previous epidemiological associations between coping profile, stress, and presenteeism and were measured at baseline. This included demographic information: age (years), sex (male, female), company, form of employment (standard and non-standard), work shift (regular and shiftwork), overtime hours/month (<45 h, 45–80 h, ≥80 h), and actual rest days/month (<4 d, 4–7 d, 8–11 d, ≥12 d). Sleep duration (<5 h, 5–6 h, 6–7 h, 7–8 h, ≥8 h), exercise habits (yes/no), current smoking status (yes or no), and alcohol consumption (daily, sometimes, or never) were all considered lifestyle habits.

### 2.4. Data Analysis

Statistical analyses were conducted using Stata 17.0 (StataCorp (College Station, TX, USA) for Windows). First, participants’ characteristics and descriptive information at baseline were analyzed. 

Second, we used bivariate and multivariate regression models with a continuous outcome of presenteeism at followup. We used the two sets of explanatory variables separately for regression analyses: (1) Brief COPE’s three subscales (problem-focused, emotion-focused, and dysfunctional coping profiles) and (2) Brief COPE’s subscales for 14 types of coping profiles (active coping, instrumental support, planning, acceptance, emotional support, humor, positive reframing, religion, behavioral disengagement, denial, self-distraction, self-blame, substance use, and venting). Multivariate regression analyses were carried out, adjusted for presenteeism at baseline, age, sex, company, employment form, work shift, hours of overworking/month, rest days/month, alcohol status, smoking status, exercise habits, sleep duration, and perceived stress at baseline [7,35,36,37]. Potential multicollinearity of the adjusted variables was tested with collinearity diagnostics (variance inflation factors), which was less than 10 for all variables [38]. We also conducted Pearson’s correlation tests; the coefficients were less than 0.3 for all variables. To handle missing values, multiple imputation was used with the Multiple Imputation by Chained Equations (MICE) approach to produce 20 imputed datasets [39]. Complete data (age, sex, company, and work performance) were used to predict the missing values (employment form, employment status, shift work, overworking hours/month, actual rest days/month, alcohol status, smoking status, exercise habits, and sleep duration at baseline). For the imputations, binary variables were imputed using logistic regression, and continuous variables were imputed using a regression model. Each variable was used as a response, with the other variables serving as explanatory variables. Sensitivity analyses were performed for the above two multivariate logistic regressions using a complete-case dataset. 

Third, to examine the mediating effects of coping profiles between distress in T1 and work performance at T2, the mediation analyses were performed using the Stata command med4way (Figure 1 H2) [40]. The med4way command reported the overall association (total effect) and the four individual components: controlled direct effect (neither mediation nor interaction), reference interaction (interaction only), mediated interaction (mediated interaction), and pure indirect effect (mediation only) [40]. If the pure indirect effect was statistically significant, this indicated a potential mediating effect of coping profiles. In this analysis, the outcome was presenteeism at followup, the explanatory variable was perceived stress at baseline, or the mediators were coping profiles. We analyzed separate models for each mediator (three subscales and 14 coping profiles of Brief COPE). Presenteeism at followup, other coping profiles, age, sex, company, employment form, work shift, hours of overworking/month, rest days/month, alcohol status, smoking status, exercise habits, and sleep duration were adjusted for in the analyses.

## 3. Results

### 3.1. Participants

At baseline, 1946 of the 2137 employees participated in the study. We excluded participants with missing data on age, sex, or work performance (*N =* 128). Thus, a total of 1818 employees were included in the study at baseline; of these, 1383 responded to the follow-up survey, but 24 of these surveys were excluded because data on work performance were missing. Thus, the final number of observations was 1359 (retention rate = 69.8%).

### 3.2. Descriptive Statistics

Table 1 shows the participants’ characteristics. The overall average age was 41.2 ± 11.2 years. The average score of presenteeism was 54.3 ± 17.9. Most participants reported standard forms of employment (87.9%) and were non-shift workers (57.8%).

### 3.3. Association between Coping Profiles and Presenteeism

Figure 2 shows the association between three coping subscales at baseline and work performance at follow-up using the regression models. In the multivariate regression analysis, problem-focused coping (β = 0.72 [95% CI, −0.53 to 1.97], *p* = 0.257) was not significantly associated with work performance, whereas dysfunctional coping (β = −1.17 [95% CI, −2.28 to −0.06], *p* = 0.039) was inversely associated with work performance. Emotion-focused coping (β = −0.41 [95% CI, −1.78 to 0.95], *p* = 0.551) was not significantly associated with work performance in bivariate and multivariate regression analysis. In the complete case analysis, these three coping subscales were not significantly associated with work performance, but the trends were similar with multiple imputation models (Appendix A).

Table 2 shows the association between coping profiles at baseline and work performance at followup using bivariate and multivariate regression analysis. In the multivariate regression analysis, planning (β = 0.86 [95% CI, 0.07–1.66], *p* = 0.034) was significantly associated with work performance, whereas self-blame (β = −1.22 [95% CI, −1.83 to −0.61], *p* < 0.001) was inversely associated with work performance. Other coping profiles were not significantly associated with work performance. In the complete case analysis, planning was significantly associated with work performance and positive reframing and self-blame were inversely associated work performance (Appendix A).

### 3.4. Mediating Effects of Coping Profiles

The effects were decomposed into total effects (i.e., effects of distress at T1 on work performance at T2), controlled direct effects (i.e., effects of distress at T1 on work performance at T2 that were not explained by the mediators), reference interaction (i.e., effects of distress at T1 on work performance at T2 owing to the interaction with the mediators), mediated interaction (i.e., effects of distress at T1 on work performance at T2 owing to both the mediation and interaction with the mediators), and pure indirect effects (i.e., mediation effects). Table 3 shows the output of the mediation analysis of each coping profile on work performance. Regarding these three coping subscales, the total effect was inversely significant, indicating that stress at T1 was inversely associated with work performance at T2. The total effects were also partially explained by the controlled direct effect. Regarding problem-focused and emotion-focused coping, the three middle components of the analysis (reference interaction, mediated interaction, and pure indirect effects) were not statistically significant, and the proportion attributable to mediation among the four components was 3.5% (problem-focused) and 2.5% (emotion focused), respectively. However, dysfunctional coping profiles significantly mediated the association between distress in T1 and work performance at T2; the proportion mediated was 22.8%.

Regarding these 14 coping profiles, the total effect was negative and statistically significant, indicating that stress at T1 was inversely associated with work performance at T2. The two middle components of the analysis (reference interaction and mediated interaction) were not statistically significant in 14 coping profiles. The relatively large proportion of mediated coping between distress in T1 and work performance at T2 was 27.5% (self-blame), 6.5% (positive reframing), 4.5% (behavioral disengagement), and 3.5% (substance use). These results suggest that self-blame has an inverse mediating effect between distress in T1 and work performance at T2, whereas positive reframing may have a buffering effect between both relationships.

## 4. Discussion

To the best of our knowledge, this is the first study to investigate the association between coping profiles and work performance among Japanese employees. Our hypothesis that coping profiles would negatively affect work performance was partially supported by our findings. However, problem- and emotion-focused coping were not significantly associated with work performance. The main findings were as follows: (1) the dysfunctional coping profile was negatively associated with work performance; (2) the planning coping profile was positively associated with work performance, whereas self-blame was negatively associated with work performance; and (3) dysfunctional coping, specifically, and self-blame, mediated the association between stress and work performance. 

Consistent with our findings, some cross-sectional studies have reported that coping profiles are associated with work performance [25,26,27,41,42]. For example, a large cross-sectional study of the Canadian community showed that the dysfunctional coping profile, including drinking, smoking, and substance use, tended to decrease work performance for both sexes [42]. These findings suggest that dysfunctional coping profiles may be one of the key factors contributing to presenteeism. 

Previous studies have shown that problem-focused coping profiles are positively associated with work performance, which is inconsistent with our findings [24,42,43]. However, few studies have investigated the effectiveness of specific coping profiles within the problem-focused coping profile. Our study showed that only planning was positively associated with work performance. Similar to our study, seeking social support and active action have been shown to improve work performance, which included quantity of work, job knowledge, attendance, and building relationships [42]. 

Previous research has provided some insights into the mechanisms underlying the causal relationship between coping profiles and work performance. Brown et al. reported that work performance can be diminished by negative work events that disrupt goal-oriented behavior [24]. However, at the same time, negative work events evoke coping efforts, and coping profiles can mitigate the effects of negative emotions on performance, with some coping profiles buffering emotions and others amplifying them [24]. Our results suggest that employees who feel less distress may be better able to adopt planning as a coping profile and, in turn, improve their work performance. In contrast, employees who feel high stress tend to employ dysfunctional coping profiles, such as self-blame, resulting in low work performance. The harmful negative emotions associated with self-blame are thought to hinder rational attempts to improve work performance. Meanwhile, emotion-focused coping profiles may produce positive effects in one situation and negative effects in another. Thus, occupational health practitioners can provide the necessary resources to identify the most suitable coping profiles for their wellbeing.

This study has several limitations. First, we did not have access to complete information about the employees’ history of physical and mental health to examine how these factors may have influenced work performance [44]. Future studies need to evaluate the history of participants who have been treated for physical or mental health conditions. Second, there is the possibility of reporting bias because this study used self-administered questionnaires. Future research should include objective measures of time management data for each employee. Third, the sampling of participants might have had a selection bias. The majority of this cohort comprised male employees, and the sex distribution was not representative of the general population. Thus, the results may not be generalizable to all Japanese employees. Further, the dropped-out employees could have had more presenteeism at follow-up, resulting in an underestimation of the prospective effects of the coping profiles on work performance. Fourth, we could not obtain information on job type, length of service, and socioeconomic status—a coping profile may relate to them. Thus, future studies need to survey detailed participant information.

Despite the above limitations, the strengths of the present study include accounting for a variety of occupations, a longitudinal design, and the use of well-validated measures for presenteeism, coping profiles, and perceived stress. Additionally, some statistical models were used to investigate the association between coping profiles and work performance. From a public health perspective, this study has key implications for preventive interventions. Our findings indicate that integrating functional stress coping, increasing the use of adaptive coping, and reducing the use of maladaptive coping in a company’s stress management program may help reduce worker stress and improve work performance. However, teaching adaptive/maladaptive coping skills is not enough. Coping is not only a response to stressful experiences but also the result of coping resources [45]. Thus, creating necessary resources to increase the use of adaptive coping profiles in the workplace is indispensable for improving work performance.

## 5. Conclusions

This longitudinal study explored the relationship between stress, coping profiles, and work performance among Japanese employees. The results suggest that dysfunctional coping profiles are associated with work performance independent of stress.

## Figures and Tables

**Figure 1 ijerph-19-04806-f001:**
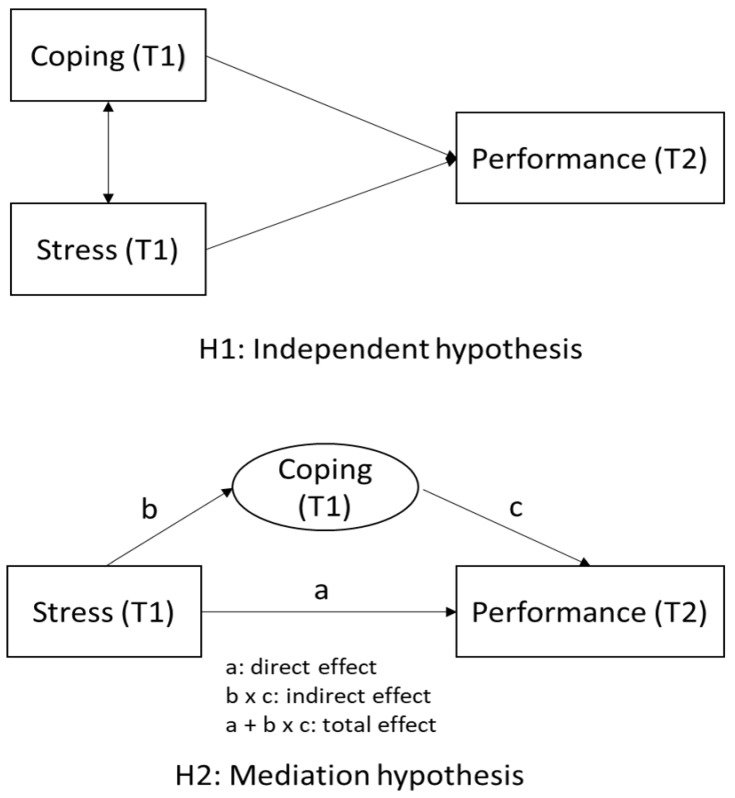
Hypotheses for the relationship between coping profiles and work performance. H2: Mediation hypothesis: “a” represents the association of Stress at T1 and work performance at T2, “b” represents the association of Stress at T1 and coping profiles at T1, and “c” represents the association of coping at T1 (M) and work performance at T2.

**Figure 2 ijerph-19-04806-f002:**
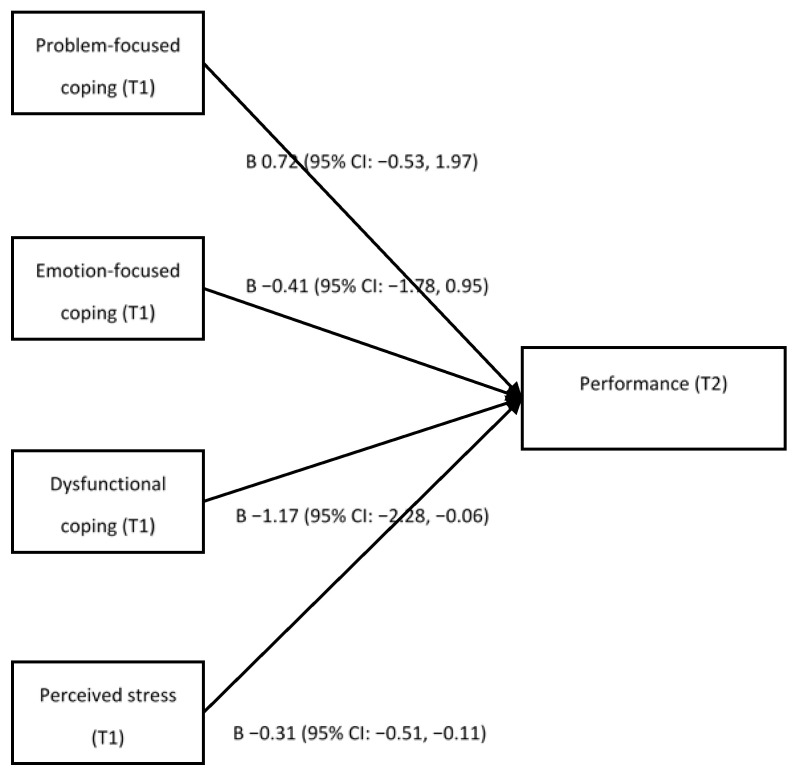
Multivariate logistic regression results for the relationship between three coping subscales and work performance, where coefficients and 95% CI are shown, adjusted for work performance, age, company, employment form, employment status, shift work, overworking hours/month, actual rest days/month, alcohol status, smoking status, exercise habits, and sleep duration at baseline; CI = confidence interval.

**Table 1 ijerph-19-04806-t001:** Participants’ characteristics at baseline (*N =* 1359).

Variables	*n*	%	Mean	(Range)	SD
Age (years)	1359		41.1	(18–66)	11.2
Sex, male	1034	76.1			
Work performance	1341		54.3	(0–100)	17.9
Problem-focused	1320		15.2	(6–24)	3.8
Active coping	1332		5.3		1.4
Use of instrumental support	1330		4.7		1.7
Planning	1334		5.2		1.6
Emotion-focused	1309		20.6	(10–40)	4.9
Acceptance	1334		5.4		1.5
Use of emotional support	1332		4.1		1.6
Humor	1328		3.8		1.6
Positive reframing	1331		4.5		1.5
Religion	1331		2.7		1.1
Dysfunctional	1293		22.7	(12–48)	5.6
Behavioral disengagement	1331		3.6		1.4
Denial	1327		2.7		1.2
Self-distraction	1335		4.7		1.6
Substance use	1327		3.3		1.8
Self-blame	1331		4.3		1.7
Venting	1331		4.1		1.5
PSS-10 score	1294		18.4	(4–37)	4.9
Form of employment					
Standard	1194	87.9			
Non-standard	158	11.6			
Unknown	3	0.2			
Work shift					
Non-shiftwork	785	57.8			
Shiftwork	571	42.0			
Unknown	3	0.2			
Overworking hours per month					
<45 h	1169	86.0			
45–80 h	133	9.8			
≥80 h	10	0.7			
Unknown	47	3.5			
Number of days off/month					
<4 days	185	13.6			
4–7 days	337	24.8			
8–11 days	683	50.3			
≥12 days	115	8.5			
Unknown	39	2.9			
Sleep duration					
<5 h/day	186	13.7			
5–6 h/day	376	27.7			
6–7 h/day	644	47.4			
7–8 h/day	87	6.4			
≥8 h/day	36	2.6			
Unknown	30	2.2			
Exercise habits (>1 h/week)					
Yes	1076	79.2			
No	251	18.5			
Unknown	32	2.4			
Current smoking (%)					
Yes	921	67.8			
No	411	30.2			
Unknown	27	2.0			
Alcohol use					
Daily	358	26.3			
Sometimes	463	34.1			
Never	513	37.7			
Unknown	25	1.8			

PSS-10: Perceived Stress Scale.

**Table 2 ijerph-19-04806-t002:** Association between coping profiles at baseline and work performance at followup using regression models.

	Bivariate Regression Analysis			
	β	95% CI	t-Value	*p*-Value
Brief COPE				
Problem-focused				
Active coping	0.41	(−0.40, 1.23)	0.99	0.322
Use of instrumental support	−0.48	(−1.22, 0.27)	−1.25	0.211
Planning	1.47	(0.64, 2.30)	3.47	0.001
Emotion-focused				
Acceptance	0.02	(−0.73, 0.77)	0.05	0.960
Use of emotional support	0.62	(−0.16, 1.40)	1.56	0.119
Humor	−0.42	(−1.10, 0.26)	−1.21	0.225
Positive reframing	−0.45	(−1.26, 0.37)	−1.08	0.281
Religion	0.45	(−0.42, 1.31)	1.02	0.310
Dysfunctional				
Behavioral disengagement	−0.66	(−1.41, 0.10)	−1.71	0.087
Denial	−1.20	(−2.09, −0.32)	−2.67	0.008
Self-blame	−1.33	(−1.96, −0.70)	−4.14	<0.001
Self-distraction	0.03	(−0.61, 0.68)	0.10	0.917
Substance use	0.22	(−0.31. 0.75)	0.82	0.412
Venting	−0.13	(−0.85, 0.58)	−0.36	0.717
PSS-10 score	−0.36	(−0.58, −0.15)	−3.37	0.001
	Multivariate regression analysis			
	β	95% CI	t-value	*p*-value
Brief COPE				
Problem-focused				
Active coping	−0.21	(−1.00, 0.58)	−0.53	0.598
Use of instrumental support	0.18	(−0.53, 0.88)	0.49	0.625
Planning	0.86	(0.07, 1.66)	2.12	0.034
Emotion-focused				
Acceptance	−0.09	(−0.80, 0.61)	−0.26	0.792
Use of emotional support	0.53	(−0.22, 1.28)	1.39	0.165
Humor	−0.15	(−0.79, 0.50)	−0.44	0.658
Positive reframing	−0.72	(−1.49, 0.04)	−1.85	0.064
Religion	0.44	(−0.39, 1.27)	1.03	0.302
Dysfunctional				
Behavioral disengagement	−0.21	(−0.94, 0.52)	−0.56	0.575
Denial	−0.63	(−1.48, 0.23)	−1.44	0.149
Self-blame	−1.22	(−1.83, −0.61)	−3.92	<0.001
Self-distraction	0.30	(−0.33, 0.94)	0.95	0.344
Substance use	0.32	(−0.25, 0.90)	1.11	0.269
Venting	−0.31	(−0.98, 0.35)	−0.93	0.353
PSS-10 score	−0.27	(−0.47, −0.07)	−2.63	0.009

PSS-10: Perceived Stress Scale; β were adjusted for work performance, age, company, employment form, employment status, shift work, overworking hours/month, actual rest days /month, alcohol status, smoking status, exercise habits, and sleep duration at baseline; CI = confidence interval.

**Table 3 ijerph-19-04806-t003:** Decomposition of the effect of each coping profile on work performance.

Potential Mediators	Total Effect		Controlled Direct Effect		Reference Interaction		Mediated Interaction		Mediation	Pure Indirect Effect	
Problem-focused	−2.71	(−3.72, −1.69)	−2.59	(−3.60, −1.58)	−0.02	(−0.06, 0.02)	0.04	(−0.03, 0.11)	3.50%	−0.13	(−0.26, 0.00)
Active coping	−2.22	(−3.27, −1.16)	−2.22	(−3.27, −1.16)	0.00	(−0.02, 0.03)	−0.01	(−0.06, 0.04)	0.10%	0.00	(−0.07, 0.08)
Use of instrumental support	−2.25	(−3.30, −1.20)	−2.24	(−3.29, −1.19)	−0.01	(−0.05, 0.03)	0.02	(−0.03, 0.07)	0.10%	−0.02	(−0.07, 0.03)
Planning	−2.20	(−3.25, −1.14)	−2.15	(−3.21, −1.09)	−0.01	(−0.03, 0.02)	0.01	(−0.02, 0.04)	1.80%	−0.05	(−0.14, 0.04)
Acceptance	−2.24	(−3.29, −1.18)	−2.24	(−3.30, −1.18)	0.00	(−0.02, 0.03)	0.00	(−0.02, 0.01)	0.10%	0.00	(−0.02, 0.02)
Emotion-focused	−2.60	(−3.60, −1.61)	−2.61	(−3.62, −1.60)	−0.03	(−0.09, 0.02)	0.07	(−0.04, 0.19)	2.50%	−0.03	(−0.23, 0.16)
Use of emotional support	−2.21	(−3.26, −1.16)	−2.21	(−3.26, −1.16)	0.00	(−0.04, 0.04)	0.01	(−0.03, 0.05)	0.20%	−0.02	(−0.08, 0.05)
Humor	−2.21	(−3.26, −1.17)	−2.22	(−3.27, −1.17)	−0.03	(−0.09, 0.02)	0.08	(−0.01, 0.17)	1.90%	−0.03	(−0.15, 0.08)
Positive reframing	−2.09	(−3.13, −1.05)	−2.22	(−3.27, −1.16)	−0.01	(−0.06, 0.03)	0.04	(−0.06, 0.14)	6.50%	0.10	(−0.06, 0.26)
Religion	−2.16	(−3.22, −1.11)	−2.11	(−3.17, −1.05)	0.01	(−0.03, 0.06)	−0.04	(−0.12, 0.03)	3.00%	−0.02	(−0.09, 0.05)
Dysfunctional	−3.38	(−4.32, −2.43)	−2.62	(−3.63, −1.60)	0.01	(−0.16, 0.18)	−0.02	(−0.34, 0.30)	22.80%	−0.75	(−1.18, −0.31)
Behavioral disengagement	−2.30	(−3.35, −1.26)	−2.20	(−3.26, −1.15)	0.01	(−0.06, 0.07)	−0.01	(−0.11, 0.09)	4.50%	−0.09	(−0.25, 0.06)
Denial	−2.40	(−3.46, −1.34)	−2.40	(−3.46, −1.34)	0.04	(−0.03, 0.12)	−0.04	(−0.12, 0.04)	1.80%	0.00	(−0.04, 0.04)
Self-distraction	−2.22	(−3.27, −1.17)	−2.22	(−3.27, −1.17)	0.00	(−0.02, 0.01)	0.01	(−0.02, 0.03)	0.10%	−0.01	(−0.05, 0.03)
Substance use	−2.15	(−3.19, −1.10)	−2.16	(−3.22, −1.11)	−0.06	(−0.13, 0.01)	0.09	(−0.01, 0.20)	3.50%	−0.02	(−0.15, 0.12)
Self-blame	−3.08	(−4.10, −2.07)	−2.22	(−3.28, −1.16)	−0.02	(−0.15, 0.11)	0.03	(−0.21, 0.28)	27.50%	−0.88	(−1.30, −0.46)
Venting	−2.23	(−3.28, −1.18)	−2.20	(−3.26, −1.15)	0.01	(−0.02, 0.04)	−0.01	(−0.06, 0.03)	1.40%	−0.02	(−0.09, 0.05)

Values were presented in coefficients and 95% confidence interval. Mediation represented proportion mediated (pure indirect effects) among total effect, adjusted for age, company, job type, employment status, shift work, overtime hours/month, actual rest days/month, alcohol consumption, smoking status, exercise habits, sleep duration, work performance, and other coping profiles at baseline.

## Data Availability

The data presented in this study are available on request from the corresponding author. To protect the privacy, data are not publicly available.

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
