# Peer review of "Associations between Coping Profile and Work Performance in a Cohort of Japanese Employees"

_ijerph, 2022, doi:10.3390/ijerph19084806_

Round 1
Reviewer 1 Report
This is an interesting study that focused on the predictive role of coping to later job stress among a large sample of workers, and most of the problems I found in the first version were improved in the second version. However, as the results of revision, some new questions have emerged as follows. Generally saying, the description of analysis procedure is premature and I cannot understand what the authors did. As a result, I cannot fully understand the results. They should be resolved before we decide to accept the article for Int J Env Res Pub Health.
[1] As already mentioned, the authors should clarify the difference between coping strategy and coping profile. According to many researchers, coping profile is not merely the combination of coping strategies. If I choose some coping strategies in a specific situation, the combination is not coping profile but coping strategies (compare this with Line 56-57). If I often talk about personal problems with my friends, this is coping profile. As far as I understand, the authors assessed coping profile in the present study, using COPE. Even if the authors use other definition for these words, the terminology (i.e. strategies or profile) in this article appears inconsistent. Please examine your context carefully and give appropriate definition.
[2] The authors illustrated two models in Line 78-86 and Figure 1.
- What is “Predictor stress” for H1 in Figure 1? What is the difference between this and “Stress” for H2? Do you mean “Stress” is not a predictor in H2?
- Please clarify the meaning of mediating (Line 83) or mediation (Figure 1), compared with H1, not in statistical words, but in theoretical words on coping. In other words, show the reason wht you examined the two hypotheses. The explanation should correspond to the statistical methods shown in Data Analysis section.
[3] Please fully describe the manipulation for COPE scores. I suppose, for example, the authors summed up three subscale scores (active coping, instrumental support, and planning) to calculate problem-focused coping scores and, then, convert them into z-scores. However, the above manipulation is not described in Line 138-140.
[4] Please clarify the purpose to do chi-square test and t-test (Line 165-167). Where are the results of these tests?
[5] Description for regression analysis (Line 165-192) is premature and makes me confused. I think the authors made two multivariate analyses, corresponding to H1 and H2, respectively. However, I cannot understand how the description in Line 167-175/182-192 corresponds to the two models.
A) It is one of the reasons for my confusion that the authors inserted the description for missing values (Line 176-182) between the description for two main analyses. Anyway, please specify the variables to which MICE was applied (Line 176-178). I think MICE was not applied to the age, sex, and work performance, because the participants with no data for them were excluded for the analysis (Line 196-197). In addition, “and work performance” (Line 197) should be changed into “or work performance”.
B) Anyway, please clarify how did you deal with independent variables. For example, in Line 169-170, three independent variables (problem-focused, emotion-focused, and dysfunctional coping) appear to be used in a multiple regression model; is this reported in Figure 2? Then, I cannot understand what is “all models in Line 170. Furthermore, in Table 2, 13 coping scores were separately used for analysis. These descriptions are not consistent. The difference between Figure 2 and Table 2, and that between “crude model” and “multivariate mode” should be clarified in Data Analysis section.
C) As for the case of mediation analysis (Line 184-192), please specify independent (exposure) variables and mediation variables. Line 189-192 does not mean these variables were used as independent variables. It is unclear whether three mediation variables in Table 3 were analyzed separately or simultaneously. In addition, it is also unclear why the authors duplicated the analysis in Table 3 and that in Table 4. The results in Table 4 include those in Table 3, and those in Table 3 include those in Figure 3. In short, the description for what the authors did is redundant and incomplete.
D) “Crude model” (Line 168) is in question, if beta in the upper half of Table 2 are adjusted for work performance etc. at baseline. If not adjusted, I recommend “bivariate regression analysis” instead of “crude regression analysis” (Line 168). Anyway, I cannot understand the difference between the upper half and the lower half of Table 2, based on the description in Line 167-176. If you adjusted the effects of work performance etc. for both, both are multivariate models. Does the footnote of Table 2 concern only with the lower half of Table 2?
E) “Exposure variables” (Line 169) is premature naming, although this is often used in epidemiologic literatures, and is not incorrect. Since coping is not an environmental factor, it should be changed into “explanatory variables”, “predictive variables”, or “independent variables”.
F) The authors calculated Spearman’s correlation coefficients (Line 175). This nonparametric analysis is inconsistent with the fact that you have already chosen a parametric method (regression analysis). You should use Pearson’s correlation coefficients.
[6] Although retention rate (69.8%) at T2 is reported in Line 200, the final N of observations (=1359) is not shown here. This should be reported in the same paragraph.
[7] The authors mentioned a previous literature in Line 205-207, although I cannot understand the sentence. Anyway, this should be moved to Discussion section.
[8] In 3.3 section, “positively” (Line 214) probably should be changed into “significantly”. What are arcs and arrows among four boxes (independent variables) in Figure 2? Please clarify the meaning of “both models”, “similar trends”, and “complete case analysis” in Line 218. In Figure 2, if -0.53 and 1.97 are lower limit and upper limit of CI for problem-focused coping, respectively, this should be clarified in footnote.
[9] Please clarify the meaning of “negative” in Line 265-276; not significant, or inverse? What are major coping strategies (Line 273)? What are percentages in Figure 3? The same results seem to be duplicated in Table 3. What are the values such as -2.71 in Table 3? As is the case of Table 4. The authors should not omit the explanation.
[10] Discussion section, particularly the review of previous researcher, is redundant. Please reduce words.
[11] The association of coping profile with job stress may depend on job type, e.g. white-collar or blue-collar, clerks or researchers. Is it impossible to analyze this in the present data? This should be discussed a little more.
Author Response
Responses to Reviewer #1
We wish to express our appreciation to the reviewer for their insightful comments. They have helped us significantly improve our manuscript.
Comment 1. As already mentioned, the authors should clarify the difference between coping strategy and coping profile. According to many researchers, coping profile is not merely the combination of coping strategies. If I choose some coping strategies in a specific situation, the combination is not coping profile but coping strategies (compare this with Line 56-57). If I often talk about personal problems with my friends, this is coping profile. As far as I understand, the authors assessed coping profile in the present study, using COPE. Even if the authors use other definition for these words, the terminology (i.e. strategies or profile) in this article appears inconsistent. Please examine your context carefully and give appropriate definition.
Response: Thank you very much for your helpful comments. According to your suggestion, we have revised to clarify the difference between coping strategies and coping profile. However, the literature on coping strategies is included to facilitate the discussion on the chosen variables of the study (please see marked file).
Comment 2. The authors illustrated two models in Line 78-86 and Figure 1.
What is “Predictor stress” for H1 in Figure 1? What is the difference between this and “Stress” for H2? Do you mean “Stress” is not a predictor in H2?
Please clarify the meaning of mediating (Line 83) or mediation (Figure 1), compared with H1, not in statistical words, but in theoretical words on coping. In other words, show the reason wht you examined the two hypotheses. The explanation should correspond to the statistical methods shown in Data Analysis section.
Response: We apologize for our unclear description of predictor stress. There were no differences in stress between H1 and H2. Thus, we have corrected the term. It is not well-known whether coping affects work performance independently of stress, or whether coping mediates the relationship between stress and performance. We used multivariate logistic regression models to explain the direct predictive relationship of coping (problem-focused and emotion-focused) on work performance. Furthermore, we used causal mediation analysis to calculate the proportion explained by the mediating effect of each coping between them (Please see marked lines 79–89, and Figure 1).
Comment 3. Please fully describe the manipulation for COPE scores. I suppose, for example, the authors summed up three subscale scores (active coping, instrumental support, and planning) to calculate problem-focused coping scores and, then, convert them into z-scores. However, the above manipulation is not described in Line 138-140.
Response: Thank you for your valuable comment. We have revised the manipulation for these coping subscales (Please see page 4, lines 140–141).
Comment 4. Please clarify the purpose to do chi-square test and t-test (Line 165-167). Where are the results of these tests?
Response: Thank you for your comment. As the reviewer pointed out, for the data, a baseline statistical analysis was not required; therefore, we have removed the chi-square test and t-test from the methods and analysis sections.
Comment 5. Description for regression analysis (Line 165-192) is premature and makes me confused. I think the authors made two multivariate analyses, corresponding to H1 and H2, respectively. However, I cannot understand how the description in Line 167-175/182-192 corresponds to the two models.
Response: Thank you for your comment. We have revised the data analysis section (Please see page 5, lines 170–203).
- A) It is one of the reasons for my confusion that the authors inserted the description for missing values (Line 176-182) between the description for two main analyses. Anyway, please specify the variables to which MICE was applied (Line 176-178). I think MICE was not applied to the age, sex, and work performance, because the participants with no data for them were excluded for the analysis (Line 196-197). In addition, “and work performance” (Line 197) should be changed into “or work performance”.
Response: Thank you for your comment. We have specified the variables to which MICE was applied (Please see page 5, lines 182-185).
B) Anyway, please clarify how did you deal with independent variables. For example, in Line 169-170, three independent variables (problem-focused, emotion-focused, and dysfunctional coping) appear to be used in a multiple regression model; is this reported in Figure 2? Then, I cannot understand what is “all models in Line 170. Furthermore, in Table 2, 13 coping scores were separately used for analysis. These descriptions are not consistent. The difference between Figure 2 and Table 2, and that between “crude model” and “multivariate mode” should be clarified in Data Analysis section.
Response: Thank you for your helpful recommendation. We have revised these points (Please see page 5, lines 170–177).
C) As for the case of mediation analysis (Line 184-192), please specify independent (exposure) variables and mediation variables. Line 189-192 does not mean these variables were used as independent variables. It is unclear whether three mediation variables in Table 3 were analyzed separately or simultaneously. In addition, it is also unclear why the authors duplicated the analys is in Table 3 and that in Table 4. The results in Table 4 include those in Table 3, and those in Table 3 include those in Figure 3. In short, the description for what the authors did is redundant and incomplete.
Response: Thank you for your helpful recommendation. We have specified the independent and mediation variables. We analyzed them separately and revised these points (Please see page 5, lines 191–203). We have merged Table 3 and 4 to Table 3.
D) “Crude model” (Line 168) is in question, if beta in the upper half of Table 2 are adjusted for work performance etc. at baseline. If not adjusted, I recommend “bivariate regression analysis” instead of “crude regression analysis” (Line 168). Anyway, I cannot understand the difference between the upper half and the lower half of Table 2, based on the description in Line 167-176. If you adjusted the effects of work performance etc. for both, both are multivariate models. Does the footnote of Table 2 concern only with the lower half of Table 2?
Response: Thank you for your comments. We have revised to change from “crude regression analysis” to “bivariate regression analysis”. In addition, we have added the explanation in the footnote (Please see page 5, line 169)..
E) “Exposure variables” (Line 169) is premature naming, although this is often used in epidemiologic literatures, and is not incorrect. Since coping is not an environmental factor, it should be changed into “explanatory variables”, “predictive variables”, or “independent variables”.
Response: Thank you for your recommendation. We have changed “Exposure variables” to “Explanatory variables” (Please see page 5, line 170)...
F) The authors calculated Spearman’s correlation coefficients (Line 175). This nonparametric analysis is inconsistent with the fact that you have already chosen a parametric method (regression analysis). You should use Pearson’s correlation coefficients.
Response: We have removed the Spearman’s correlation coefficient and replaced it with Pearson’s correlation coefficient. The results were similar between Pearson and Spearman’s correlation (Please see marked page 5, line 180).
Comment 6. Although retention rate (69.8%) at T2 is reported in Line 200, the final N of observations (=1359) is not shown here. This should be reported in the same paragraph.
Response: Thank you for your recommendation. We have added the final number of observations (Please see page 5, line 210).
Comment 7. The authors mentioned a previous literature in Line 205-207, although I cannot understand the sentence. Anyway, this should be moved to Discussion section.
Response: Thank you for pointing out this discrepancy. We have deleted this sentence.
Comment 8. In 3.3 section, “positively” (Line 214) probably should be changed into “significantly”. What are arcs and arrows among four boxes (independent variables) in Figure 2? Please clarify the meaning of “both models”, “similar trends”, and “complete case analysis” in Line 218. In Figure 2, if -0.53 and 1.97 are lower limit and upper limit of CI for problem-focused coping, respectively, this should be clarified in footnote.
Response: Thank you for this comment. We have carefully revised our manuscript according to the suggested change. The arcs and arrows between the four boxes (independent variables) in Figure 2 mean that we have performed a multiple logistic analysis. However, these have been removed as they may be misleading to the readers (Please see Figure 2 and lines 219–244).
Comment 9. Please clarify the meaning of “negative” in Line 265-276; not significant, or inverse? What are major coping strategies (Line 273)? What are percentages in Figure 3? The same results seem to be duplicated in Table 3. What are the values such as -2.71 in Table 3? As is the case of Table 4. The authors should not omit the explanation.
Response: Thank you for your recommendation. We have revised and added explanations for Table 3 and Table 4. In addition, we have omitted Figure 3 due to the duplicated Table 3. (Please see lines 252–277).
Comment 10. Discussion section, particularly the review of previous researcher, is redundant. Please reduce words.
Response: Thank you for your valuable comment. We have revised the discussion section to reduce the review of previous studies.
Comment 11. The association of coping profile with job stress may depend on job type, e.g. white-collar or blue-collar, clerks or researchers. Is it impossible to analyze this in the present data? This should be discussed a little more.
Response: Thank you for your valuable comment. We could not distinguish these job types. Thus, we have added these points under the limitations (Please see lines 332–344).
Thank you again for your feedback on our paper. We hope that the revised manuscript is now suitable for publication.
Reviewer 2 Report
The authors revised the manuscript and corrected or added the text which was needed for better understanding of research results. I have no comments.
Author Response
Thank you for your feedback on our paper. We hope that the revised manuscript is now suitable for publication.
Reviewer 3 Report
First of all, thank you for being able to revisit this article. The authors have worked hard on it. Congratulations
This study aimed to investigate the effects of coping strategies on work performance. The data were collected during a prospective 2-year cohort study with 1359 employees in Japan. Participants completed a self-administered questionnaire in 2018 (T1; baseline) and again in 2020 (T2; follow-up; follow-up rate: 69.8%) to enable the evaluation of work performance, perceived stress and coping strategies in T1 and T2. Multivariate logistic regression models and causal mediation analysis methods were performed to identify the effects of coping strategies on work performance. Covariates included age, sex, enterprise, type of work, working status, working hours, holidays, and lifestyle behaviors (e.g., smoking, length of sleep). Dysfunctional coping profile (β=-1.17 [95% CI, -2.28 to -0.06], p=0.039) was negatively associated with work performance. Planning coping strategies (β=-0.86 [95% CI, 0.07-1.66]) and autoculpability (β=-1.33 [95% CI, -1.96 to -0.70], p<0.001) were significantly associated with work performance. Dysfunctional confrontation, specifically, self-blame (β=-1.22 [95% CI, -1.83 to -0.61]), measured the association between stress and work performance. The authors conclude that some coping strategies can lead to an increase or decrease in work performance and It must be recognized the possible impact of coping strategies on the labor productivity of workers.
I think that the authors have worked a lot on the suggestions of the reviewers and have satisfied my concerns in this regard. The article can be published in its current form.
I encourage the authors to continue their research on the subject.
Kind regards
Author Response

(The authors gave the same response as above.)

Round 2
Reviewer 1 Report
After revision, this article has become much easier to be read. However, I still find some questions partially caused by the errors in English and careless mistake. Some more revision is required to be accepted as an article in Int J Env Res Pub Health.
[1] Coping strategy and coping profile are still confused.
Line 55-62 should be as follows;
When encountering stressful situations, individuals employ a combination of varied coping strategies [18]. The coping strategies can be categorized into three to four types. COPE and its shortened version, Brief-COPE, two of the most commonly used coping scales, have suggested some common patterns of strategies called coping profiles [19, 20]. For instance, some individuals tend to use a combination of problem-focused and emotion-focused coping strategies, whereas some individuals tend to employ primarily dysfunctional coping strategies, and others use very few strategies to deal with life's stressors [21-23].
[2] In Line 84, is “or” correct? Is it “and”?
[3] In the lower half of Figure 1, I hope additional explanation such as “direct effect” and “indirect effect”.
[4] Both of 14 scales and three scales are named “subscales” in previous literatures. The authors call the latter “profiles”, although this is curious in English. Line 130-139 should be as follows;
The Brief Coping Orientation to Problems Experienced (Brief-COPE) was used to evaluate coping profiles. This scale consists of 28 items that measure 14 different types of coping profile, including active coping, planning, positive reframing, acceptance, humor, religion, emotional support, instrumental support, self-distraction, denial, venting, substance use, behavioral disengagement, and self-blame [20]. Each item was rated on a 4-point Likert scale, and higher scores indicate more frequent use of a coping category. Copper et al. described the 14 subscales of the Brief-COPE as reflecting predominantly problem-focused (active coping, instrumental support, planning), emotion-focused (acceptance, emotional support, humor, positive reframing, religion), and dysfunctional coping profiles (behavioral disengagement, denial, self-distraction, self-blame, substance use, venting) [31, 32].
[5] “We analyzed the two explanatory variables separately” in Line 170 is incorrect. This should be “We used the two sets of explanatory variables separately for regression analyses”.
[6] In Line 172, “14 coping profiles” should be “subscales for 14 types of coping profiles”.
[7] In Line 174, “Multivariate regression models were adjusted for” should be “Multivariate regression analyses were carried out, adjusted for”.
[8] In Line 191, “to survey the effects of mediated coping profiles” should be “to examine the mediating effects of coping profiles”.
[9] In Line 200, “and” should be “or”.
[10] In Line 232-233, “PSS-10: Perceived Stress Scale” should be deleted.
[11] In Line 247, “Multivariate models were adjusted for” should be “β were adjusted for”.
[12] In Table 2 and 3, lower/upper limit of 95% CI should be put in parentheses.
[13] In Line 293, “self-blame,” should be “and self-blame”.
[14] Clarify whether “Similarly, seeking social support … [43]” indicates the results obtained in the present study, or results in Ref. 43.
Author Response
Responses to Reviewer #1
We wish to express our strong appreciation to the reviewers for their insightful comments on our paper. We feel the comments have helped us significantly improve the paper.
Comment 1. Coping strategy and coping profile are still confused.
Line 55-62 should be as follows;
When encountering stressful situations, individuals employ a combination of varied coping strategies [18]. The coping strategies can be categorized into three to four types. COPE and its shortened version, Brief-COPE, two of the most commonly used coping scales, have suggested some common patterns of strategies called coping profiles [19, 20]. For instance, some individuals tend to use a combination of problem-focused and emotion-focused coping strategies, whereas some individuals tend to employ primarily dysfunctional coping strategies, and others use very few strategies to deal with life's stressors [21-23].
Response: Thank you for your valuable comment. According to your suggestion, we have revised above sentences. (Please see marked lines 55-62).
Comment 2. In Line 84, is “or” correct? Is it “and”?
Response: Thank you very much for your helpful comments. We have revised from “or” to “and”. (Please see marked line 84).
Comment 3. In the lower half of Figure 1, I hope additional explanation such as “direct effect” and “indirect effect”.
Response: Thank you for your valuable comment. We have added explanation of “direct effect” and “indirect effect”. (Please see page 3, Figure 1).
Comment 4. Both of 14 scales and three scales are named “subscales” in previous literatures. The authors call the latter “profiles”, although this is curious in English. Line 130-139 should be as follows;
The Brief Coping Orientation to Problems Experienced (Brief-COPE) was used to evaluate coping profiles. This scale consists of 28 items that measure 14 different types of coping profile, including active coping, planning, positive reframing, acceptance, humor, religion, emotional support, instrumental support, self-distraction, denial, venting, substance use, behavioral disengagement, and self-blame [20]. Each item was rated on a 4-point Likert scale, and higher scores indicate more frequent use of a coping category. Copper et al. described the 14 subscales of the Brief-COPE as reflecting predominantly problem-focused (active coping, instrumental support, planning), emotion-focused (acceptance, emotional support, humor, positive reframing, religion), and dysfunctional coping profiles (behavioral disengagement, denial, self-distraction, self-blame, substance use, venting) [31, 32].
Response: Thank you for your valuable comment. According to your suggestion, we have revised above sentences. (Please see marked lines 133-143).
Comment 5. “We analyzed the two explanatory variables separately” in Line 170 is incorrect. This should be “We used the two sets of explanatory variables separately for regression analyses”.
Response: We thank the reviewer for this comment. This error has been corrected in accordance with the reviewer's comment. (Please see marked lines 174-175).
Comment 6. In Line 172, “14 coping profiles” should be “subscales for 14 types of coping profiles”.
Response: Thank you for your recommendation. We have revised this point. (Please see marked lines 176-177).
Comment 7. In Line 174, “Multivariate regression models were adjusted for” should be “Multivariate regression analyses were carried out, adjusted for”.
Response: We thank the reviewer for this comment. In accordance with the reviewer's comment, we have revised the phrase. (Please see marked lines 179-180).
Comment 8. In Line 191, “to survey the effects of mediated coping profiles” should be “to examine the mediating effects of coping profiles”.
Response: Thank you for this comment. In accordance with the reviewer's comment, we have revised the phrase. (Please see marked line 195).
Comment 9. In Line 200, “and” should be “or”.
Response: Thank you for your recommendation. In accordance with the reviewer's comment, we have revised it. (Please see line 203).
Comment 10. In Line 232-233, “PSS-10: Perceived Stress Scale” should be deleted.
Response: Thank you for your comment. We have deleted this phrase.
Comment 11. In Line 247, “Multivariate models were adjusted for” should be “β were adjusted for”.
Response: Thank you for your recommendation. We have revised this point. (Please see marked line 252).
Comment 12. In Table 2 and 3, lower/upper limit of 95% CI should be put in parentheses.
Response: Thank you for your recommendation. We have revised this point. (Please see Table 2 and 3).
Comment 13. In Line 293, “self-blame,” should be “and self-blame”.
Response: We thank the reviewer for this comment. This error has been corrected in accordance with the reviewer's comment. (Please see marked line 296).
Comment 14. Clarify whether “Similarly, seeking social support … [43]” indicates the results obtained in the present study, or results in Ref. 43.
Response: Thank you for your recommendation. We have revised this point. (Please see marked line 308).
Thank you again for your feedback on our paper. We hope that the revised manuscript is now suitable for publication.

This manuscript is a resubmission of an earlier submission. The following is a list of the peer review reports and author responses from that submission.
Round 1
Reviewer 1 Report
This article focused on the relationship between job stress and coping among workers, a very interesting and important issue in the field of occupational health. The authors carried out longitudinal study on the issue and obtained data from a large sample. However, I find some important problems as follows, and cannot confirm the article is acceptable for Int J Env Res Pub Health. The authors should examine the problems carefully to submit the paper.
The authors referred to 39 literatures in main text. However, only 37 are listed at the end of article, apparently showing something wrong. I had difficulty in tracing the literatures.
The hypotheses to be examined in this research are shown in P2. It is reasonable that the authors hypothesized problem-focused coping as a positive correlate to work performance. However, the reason why the authors hypothesized emotion-focused coping as a negative correlate to work performance is not explained.
WHO-HPQ to assess work performance is a single-item scale from 0 to 10. However, the manipulation to calculate presenteeism scores from WHO-HPQ scores are not fully demonstrated in P4 (line 96-98). How many times the authors used WHO-HPQ?
Carvres’ COPE is a popular tool to assess individual coping profile, although COPE’s factor structure is inconsistent among many researches. The brief-COPE includes 14 subscales, as shown in line 101-102, and they are independent each other. Is it permitted to summarize them into three scores (problem-focused, emotion-focused, and dysfunctional coping)? The authors should show the premise of this manipulation with literatures. Another problem on terminology should be discussed. Coping strategy means a way of coping with a specific situation or stressor, while coping profile is the coping strategies that an individual often uses. We should distinguish coping profile from coping strategies.
PSS-10 is a tool to assess strain. However, the authors simply demonstrate strain as “stress” (line 112-113). This terminology is incorrect.
Some covariates to presenteeism are considered in multivariate analyses. I suppose some of them (i.e. overtime hours and sleep duration) had internal correlation. However, I find no explanation about multiple co-linearity. The way of handling missing values is another point in question. If missing values are not so frequent, we usually exclude the cases with missing values from analysis. Why the authors used other methods?
The reason why the authors used GEE also should be discussed. We can use multiple regression analysis or multiple logistic analysis to predict work performance at T2, based on independent variables at T1. Why the authors use different analysis for the same data? The reason should be discussed in more detail in 2.4..
Reviewer 2 Report
The study is very interesting and presents the relationship between coping strategies and work performance. I have some comments and recommendations related to the manuscript.
Figure 1 shows the hypothesis (H1 and H2), however, the final relationships are were not shown in a figure. This could help to understand better the study contribution. I suggest adding a figure (or figures) where the B can be observed.
About the Brief-COPE. The methodology section mentions it includes 28 items measuring 14 strategies, and three subscales. Were the strategies analyzed as individual variables? Why coping was shown as a box and not as a circle (ellipse) in figure 1?
Work performance, coping strategies, and stress got data from a survey, as result, it is important to show reliability analysis. For PSS-10, a Cronbach´s Alpha of 0.84 was reported but not for work performance and stress. I suggest reporting these values.
Finally, although it is mentioned in the limitations, it could be interesting to see a multivariate analysis of the demographic variables and their influence on the observed variables.
Reviewer 3 Report
Well done study, well prepared manuscript. I have only one suggestion: the procedure should be described more clearly. It should be clear what instruments were used in the test I and what instruments were used in the test II. Is the stress was measured once or twice, is the instrument of coping strategies were used once or twice? Please clarify the procedure of the research in the methodology part.
Reviewer 4 Report
Good morning, first of all, thank you for the opportunity to review the manuscript,
The article addresses the associations between coping strategies and job performance in Japanese employees.
The aim of the study was to investigate the effects of coping strategies on job performance. The article concludes how the strategies can lead to an increase or decrease in work performance, the possible impact of coping strategies, the need to recognize policies on the productivity of workers' work and education on care strategies should be incorporated into occupational health examinations.
This is a good work, on a very interesting topic, but before publishing it should incorporate the following suggestions, I ask the authors to follow them one by one,
some important references are missing in the introduction, check it out,
although it is deduced from the text, the research gap that the article tries to cover must be explained,
the discussion section must be improved and more elaborated, the bibliography of the previous studies of the introduction must be connected with the discussion, that will give more power to the manuscript,
Finally, the bibliography must be ordered, following the regulations of the journal.
With these changes made, the article will improve and have the quality to be published in this prestigious journal,
Kind regards